# Short Term Effects of Proprioceptive Neuromuscular Facilitation Combined with Neuromuscular Electrical Stimulation in Youth Basketball Players: A Randomized Controlled Trial

**DOI:** 10.3390/jfmk9040280

**Published:** 2024-12-20

**Authors:** Manuel Sos-Tirado, Aser Campo-Manzanares, Lidia Aguado-Oregui, Carles Cerdá-Calatayud, Juan Carlos Guardiola-Ruiz, Celia García-Lucas, Francisco Javier Montañez-Aguilera, Juan Francisco Lisón, Juan José Amer-Cuenca

**Affiliations:** 1Department of Physiotherapy, Universidad Cardenal Herrera CEU, CEU Universities, 46115 Alfara del Patriarca, Spain; manu.sos.elians@gmail.com (M.S.-T.); asercampomanzanares@gmail.com (A.C.-M.); lidiaaguado98@gmail.com (L.A.-O.); carlescercal@gmail.com (C.C.-C.); juancarlosguardiolaruiz@gmail.com (J.C.G.-R.); francisco.monta@uchceu.es (F.J.M.-A.); juanjoamer@uchceu.es (J.J.A.-C.); 2Department of Biomedical Sciences, Universidad Cardenal Herrera CEU, CEU Universities, 46115 Alfara del Patriarca, Spain; celia.garcialucas1@uchceu.es; 3Centre of Physiopathology of Obesity and Nutrition (CIBERobn), CB06/03/0052, Instituto de Salud Carlos III, 46026 Valencia, Spain

**Keywords:** hamstring flexibility, vertical jump, young basketball players, PNF, NMES

## Abstract

**Background:** Hamstring muscle injuries are common in basketball and result in long periods of inactivity. To reduce their incidence, preventive protocols, including proprioceptive neuromuscular facilitation (PNF) stretches, have been proposed. The aim of this study is to compare the short-term effects of PNF and PNF + neuromuscular electrical stimulation (NMES) on hamstring extensibility and, secondarily, on vertical jump capacity in young basketball players. **Materials and Methods:** The study was a randomized controlled trial. One group performed a PNF stretching protocol and the other PNF + NMES. Hamstring extensibility was measured using the Sit and Reach test and the popliteal angle and jump capacity were measured using the Counter Movement Jump, both before and immediately after the intervention. **Results:** Forty-five young male players participated. Both groups showed significant intra-group improvements (*p* < 0.001) in hamstring flexibility after the intervention. However, there were no significant intra-group differences (*p* > 0.05) in jump capacity. Additionally, no significant differences (*p* > 0.05) were observed between the two groups for any of the measured variables. **Conclusions:** Both programs are effective in increasing hamstring flexibility in the short term without impairing vertical jump capacity in young basketball players.

## 1. Introduction

Basketball is a globally celebrated team sport that has seen substantial growth in youth participation across both competitive and recreational contexts in recent decades [1,2]. This dynamic sport is defined by continuous transitions between offense and defence, incorporating diverse movements and alternating periods of high and moderate intensity [3]. To excel, basketball players require endurance, speed, explosive power, and agility. Among these attributes, vertical jump capacity is particularly critical, as it underpins fundamental game actions such as driving to the basket, securing rebounds, and blocking shots [3,4,5]. Flexibility, though often overlooked, plays a vital role in improving performance by enhancing the ability to stretch or reach during sports, promoting efficient movement patterns, and reducing resistance in muscles that are more compliant or less stiff during intended movements [6].

The lower limbs face intense physical demands in basketball, making optimal conditioning in this region essential. Hamstring tightness or shortening is one of the most common musculoskeletal issues among both athletes and the general population [7], with males experiencing a higher degree of shortening [8,9]. Research indicates that hamstring tightness can lead to inefficient mobility and reduced force production due to imbalances in lower-extremity joint forces [10]. Furthermore, poor hamstring flexibility has been associated with an increased risk of strains [7,11,12,13], though the relationship remains inconclusive [14]. Hamstring strains are the fourth most frequent injury among professional basketball players, accounting for 3.1% of total days lost due to injury [15].

Flexibility is a cornerstone of physical conditioning programs, as it enables tissues to adapt to stress, enhances movement efficiency, and may reduce injury risk, particularly in high-demand sports like basketball [16]. Specifically, improving hamstring flexibility supports smoother, more efficient movements by optimizing joint mechanics and muscle extensibility, which are essential for explosive actions such as sprints and rapid directional changes. Mechanisms underlying these benefits include shifting the muscle length-torque curve, addressing shortened optimum muscle length [17,18], and mitigating the effects of synergistic dominance, as well as mitigating the effects of synergistic dominance. The latter, as described by Buckthorpe et al. [19], refers to the compensatory activation of dominant muscles over weaker ones, which can lead to imbalances and inefficiencies in joint mechanics. These principles highlight the potential importance of hamstring flexibility in supporting athletic performance in basketball.

Stretching techniques, including static stretching, dynamic stretching, and proprioceptive neuromuscular facilitation (PNF), are widely used to enhance flexibility and range of motion (ROM) [7,16]. Among these, PNF techniques are often favoured for their superior effectiveness in increasing ROM [20,21]. PNF involves alternating stretching with voluntary isometric contractions of either agonist or antagonist muscles. When the agonist muscle contraction is incorporated, the technique is referred to as contract-relax PNF (crPNF) [21,22,23], although some sources use the term “hold-relax” to emphasize its isometric nature.

In recent years, the combination of stretching with neuromuscular electrical stimulation (NMES) has gained attention for its potential to further enhance flexibility [24,25,26]. NMES is a low-frequency transcutaneous electrotherapy that activates alpha motor neurons, inducing involuntary muscle contractions [27]. Superimposing NMES during stretching is thought to amplify muscle contraction strength, potentially enhancing the inhibitory effect on the H-reflex and improving muscle extensibility. Additionally, NMES may modulate pain perception, increasing tolerance and facilitating greater gains in flexibility. Studies suggest that NMES combined with PNF yields superior ROM improvements compared to stretching alone [24,25]. However, most research on crPNF and crPNF + NMES has been conducted in general adult populations [20,22,23,24,25,28,29,30], leaving a gap in the literature regarding its application to athletic populations, particularly young male basketball players. Therefore, this study aims to compare the effectiveness of a crPNF protocol and a crPNF + NMES protocol in enhancing hamstring flexibility in this population. Additionally, we aim to evaluate whether either protocol affects vertical jump performance, a key indicator of athletic ability in basketball.

## 2. Materials and Methods

### 2.1. Study Design

The study was a randomized controlled trial (NCT06648356). The Ethics Committee for Biomedical Research of the Universidad Cardenal Herrera (Report CEEI24/531) approved the study following the Declaration of Helsinki. CONSORT criteria for randomized controlled trials were applied [31].

### 2.2. Participants

A total of 45 participants were recruited between 15 April and 28 April 2024, and the measurements of the variables were conducted between 30 April and 8 May 2024. Various coaches were contacted to recruit basketball players from the “Club Baloncesto Moncada” as study participants, where all interventions and measurements were conducted. The participants competed in regional leagues organized by the Federación de Baloncesto de la Comunidad Valenciana (FBCV). Participants who met the following inclusion criteria were selected: (1) active basketball players who train at least 3 days a week; (2) ages between 11 and 18 years; (3) male gender. The exclusion criteria were: (1) participation in an organized hamstring stretching program; (2) presence of low back pain; (3) hamstring muscle injuries in the last 6 months; (4) spinal or abdominal surgeries in the last 6 months.

### 2.3. Interventions

Informed consent from the parents or legal guardians of the volunteers was obtained before their participation in the trial, and the participants’ agreement was also recorded.

Participants were randomly assigned to one of two intervention groups.

Prior to the measurements, all participants performed a standardized 10-min warm-up to ensure consistency and reduce variability in the results. The warm-up included jogging, dynamic stretching, lower and upper limb strength exercises, submaximal plyometric exercises, and submaximal intermittent running with directional changes.

Participants in the first group (crPNF Group) performed an isolated crPNF stretching protocol. They were placed in a long sitting position with maximum knee extension possible until a moderate-strong stretch sensation was felt without pain. The stretch duration was 20 s, followed by a maximal voluntary isometric contraction (MVIC) of the hamstrings for 5 s [20]. Three stretch-contraction cycles were completed. To minimize pelvic tilting and back extensor involvement, participants were instructed to maintain a neutral spine position, which was monitored by the researchers. Isometric contraction was performed by resisting against the plinth, providing a stable point of resistance. Participants were instructed to contract at maximal voluntary effort (100% MVIC). A second researcher controlled the stretching and contraction times (Figure 1A).

Participants in the second group (NMES Group) performed the same stretching protocol [20], but with NMES (symmetrical biphasic rectangular pulse of 50 Hz and 300 µs phase width) superimposed on the isometric contraction. NMES superimposed during voluntary contractions is a proven method to enhance muscle performance without external loads [32]. The selected NMES parameters have been proved to be effective and appropriate for this purpose [32,33]. A TensMed S82 electrostimulator was used (Enraf-Nonius, Delft, The Netherlands). Before the intervention, two 5 × 9 cm electrodes (Herycor Rehabilitación y Medicina Deportiva S.L., Alicante, Spain) were placed along the hamstrings, and participants were asked to indicate the current intensity at which they felt a moderate-strong but not painful contraction. During the intervention, a first researcher maintained the stretch position, and a second researcher adjusted the current intensity and controlled the stretching and contraction times (Figure 1B).

### 2.4. Variables and Measurements

To compare the effects of the interventions, three variables were employed: hamstring flexibility assessed with the classic Sit and Reach Test (SR); hamstring flexibility assessed with the Popliteal Angle Test (PA); and jumping ability assessed with the Counter Movement Jump test (CMJ). The tests were always conducted in the same order—SR, PA, and CMJ—to ensure consistency across participants and to enhance the reproducibility of the methodology. The same blinded examiner recorded the values of each variable in each participant immediately before and immediately after the intervention to ensure its effect was not lost.

The classic Sit and Reach Test has high intra-examiner reliability and is validated for measuring hamstring extensibility [34]. For its execution, subjects were placed in a long sitting position with the soles of their feet against the base of the measurement box. Keeping the knees extended, they sought to reach the maximum possible distance in the box with their fingers (Figure 2B) [35]. The test was repeated three times, and the average was obtained. Values were recorded in cm, where a greater distance in cm indicates a higher degree of hamstring extensibility.

The Popliteal Angle Test is validated for measuring hamstring extensibility [36] and is widely recognized as one of the most effective techniques for goniometric measurements [37]. Participants were placed in the supine position. A researcher held the hip flexion at 90° and passively extended the knee until the participant felt a strong stretch without pain. To ensure consistency, hip flexion was stabilized manually by the same researcher for all participants throughout the study. The examiner recorded the knee extension degrees at that moment using an inclinometer (Fabrication Enterprises Inc., White Plains, NY, USA) (Figure 2A) [38]. Each reading was taken within 2 s per trial to minimize the duration of muscle engagement and reduce potential cumulative effects. The test was repeated three times, and the average was obtained. A full knee extension corresponded to a value of 0 degrees, and a higher number of degrees indicated greater hamstring shortening.

The Counter Movement Jump has been validated and standardized in adolescents for measuring jumping ability. The instructions for performing the jump were as follows: Start from an erect standing position with a straight torso, fully extended knees, hands on hips, and feet shoulder-width apart, maintaining this position for at least 2 s before the descent phase. During the push-off phase, perform a downward movement until the knee angle reaches approximately 90°. For the take-off phase, jump with maximal effort, and during the apex of the jump, keep the legs fully extended. The landing phase should occur with both feet together and knees fully extended (Figure 2C) [39]. The My Jump App, which has high reliability and validity for measuring vertical jump height in centimeters [40], was used on an iPhone 11 running iOS 17.4.1. The captures were made using video recorded at 1080p resolution and 60 frames per second (fps). A greater height indicates a greater jumping ability.

### 2.5. Randomization and Blinding

Participants were randomized into the two intervention groups using the stratification method. Based on the PA variable results, participants were classified according to their degree of hamstring shortening, normal (≤15°), grade I shortening (16–34°), and grade II shortening (≥35°), following the reference values established by the literature [36]. The PA test was chosen for stratification because it provides precise angular measurements specific to hamstring extensibility, minimizing the influence of variables unrelated to muscle flexibility, such as spinal or hip mobility, which could affect the results of the SR test. Each pair of new participants belonging to each shortening group was randomly assigned to the crPNF group or the NMES group.

This randomization was created by an investigator not involved in the records or interventions using computer software. The sequence remained hidden until assignment. The researchers who performed the interventions indicated to the participants which group they belonged to.

Due to the nature of the interventions, participants and researchers who conducted the interventions could not be blinded. The researcher who took the measurements of all participants was blinded to the group assignments.

### 2.6. Statistical Analysis

To achieve a statistically significant reduction of 4.5 degrees (SD = 5°) in the different variables between groups (PNF vs. NMES), with a statistical power of 80% and an α risk of 0.05, a sample size of 19 participants per group was needed. The sample size was increased by 15% to compensate for possible alterations in the statistical significance of the results due to potential dropouts during the intervention, resulting in a total of 44 participants being required.

Statistical analysis was performed on an intention-to-treat basis using IBM SPSS v. 27.0.1 (Armonk, NY, USA). Descriptive statistics for each variable were calculated, expressed as mean and standard deviation. The Shapiro–Wilk test checked if the variables in both intervention groups met the normality assumption. The Levene test checked for homogeneity of variances between the different intervention groups. The results of the Shapiro–Wilk test confirmed that the normality assumption was met for all variables (*p* > 0.05), and the Levene test demonstrated homogeneity of variances between the groups (*p* > 0.05).

An independent samples *t*-test was conducted to ensure no differences in the “height–wingspan” variable between intervention groups. To compare intra-group and inter-group differences in the different variables (SR, PA, and CMJ) before and after the intervention, a two-way ANOVA with repeated measures in one factor was used. The intervention group (PNF or NMES) was the between-subjects factor, and the measurement time (pre-intervention and post-intervention) was the within-subjects factor. Bonferroni post-hoc tests were applied following the ANOVAs. Cohen’s d effect sizes were calculated, with thresholds of 0.2, 0.5, and 0.8 corresponding to small, medium, and large effect sizes, respectively. A significance level of *p* < 0.05 was established.

To evaluate the relative reliability of the tests performed, the intraclass correlation coefficient (ICC) with its 95% confidence interval (95% CI) was calculated. Three trials were conducted for each test, with the ICC used to assess the consistency of the measurements. The ICC values were interpreted following standard classification criteria, with values above 0.9 considered indicative of excellent reliability. Additionally, absolute reliability was assessed by calculating the standard error of measurement (SEM) and the minimal detectable change at 95% confidence (MDC95).

## 3. Results

The final sample consisted of 45 basketball players with a mean age of 14.4 years (1.5 standard deviation). Table 1 shows the demographic characteristics of participants in both intervention groups.

Figure 3 shows a flow diagram of the participant recruitment process, their assignment to intervention groups, and the subsequent analysis of the results.

The stratified randomization process was considered successful as no significant between-group differences were found for any of the three dependent variables before the intervention (*p* > 0.05). Additionally, there were no significant differences in the “height–arm span” variable between the two intervention groups (*p* > 0.05). This is relevant when comparing the results of the groups, as a larger arm span could introduce a bias in the SR test.

Significant intra-group differences were observed in both the SR and PA variables after the intervention (*p* < 0.001). However, there were no significant inter-group differences when comparing the means of the SR and PA variables after the intervention (*p* > 0.05). The CMJ variable showed no significant differences in its means either inter-group or intra-group after the intervention (*p* > 0.05). No significant differences were found in the time × group interaction for any of the three variables (*p* > 0.05). Table 2 provides the Bonferroni post hoc pairwise comparisons and corresponding effect sizes.

The analysis demonstrated excellent reliability for the three variables, with ICC values of 0.975 (95% CI: 0.959–0.985) for SR, 0.990 (95% CI: 0.984–0.994) for PA, and 0.962 (95% CI: 0.939–0.978), all statistically significant (*p* < 0.001). The SEM and MDC95 values were 2.07 and 5.7 for SR, 1.31 and 3.6 for PA, and 2.53 and 7.0 for CMJ, respectively, confirming the reliability of the measurements.

## 4. Discussion

The primary objective of this study was to compare a crPNF protocol and a crPNF + NMES protocol to determine if they increase hamstring flexibility in the short term among young basketball players. The secondary objective was to assess whether either protocol negatively affected vertical jump capacity. To our knowledge, no studies to date have analysed the immediate effects of PNF stretching protocols, with or without NMES, in young athletes.

The results show that both protocols led to a significant increase in hamstring flexibility in the short term. These findings are consistent with several studies that also utilized crPNF techniques to improve hamstring flexibility [22,23,28,29,30]. The main mechanism explaining the effectiveness of crPNF is typically attributed to autogenic inhibition, also known as the inverse myotatic reflex [21,41]. This involves a muscle contraction while the muscle’s myotendinous junction is in a stretched position, stimulating the Golgi tendon organs and increasing inhibitory signals via Ib interneurons. However, recent evidence suggests that the observed improvements from stretching may also stem from a modulation of pain tolerance, a phenomenon referred to as stretch tolerance, as opposed to structural changes in muscle extensibility [42,43]. This hypothesis highlights the need for further research to clarify the underlying mechanisms and optimize the protocols for athletic populations.

While our results indicate that crPNF can effectively improve flexibility, it is essential to interpret its potential impact on injury prevention with caution. Hamstring injuries are influenced by multiple factors, including eccentric strength, pelvic control, and flexibility, as well as non-modifiable factors such as age and gender. Flexibility alone, and stretching in particular, remains an inconclusive factor for hamstring injury prevention [36,44]. Therefore, while crPNF may contribute to a broader injury prevention strategy, it should not be considered a standalone solution.

When comparing the crPNF group and the crPNF + NMES group, no significant differences were observed in any of the variables. Although NMES has been shown to enhance muscle performance in other contexts [24,25], its lack of additional benefit in this study may be attributed to factors such as the participants’ lack of familiarity with NMES or the short duration of the intervention. Some studies suggest that repeated exposure to NMES is necessary to achieve optimal results due to gradual adaptation and tolerance to the electrical stimulus [45]. This may explain why the NMES group did not outperform the crPNF group despite its theoretical advantages. Future studies could address these limitations by extending the duration of interventions and ensuring participants have sufficient time to acclimate to NMES.

The findings of this study also demonstrate that neither protocol negatively affected vertical jump performance, an important indicator of explosive power in basketball. This is consistent with previous research indicating that brief stretching protocols do not impair, and may even support, performance when incorporated into warm-up routines [21,23]. Although the potential influence of warm-up effects from repeated trials cannot be ruled out, this would affect both groups equally, as all participants followed the same testing procedures. As such, the comparative effects of the interventions can be confidently attributed to the protocols themselves rather than to confounding factors. Future studies could further refine these findings by investigating the potential long-term effects of these interventions on flexibility, performance, and injury prevention.

Finally, this study has several limitations that should be acknowledged. The sit-and-reach test was performed using a measurement tape on a table instead of a standard sit-and-reach box. Although this setup ensured consistency across all participants and minimized potential bias, it may lead to slight differences when comparing our results to studies using the standard box. Additionally, while the potential influence of warm-up effects from repeated trials cannot be entirely ruled out, these effects would have been consistent across both intervention groups due to the standardized testing procedures. Therefore, the observed differences can be confidently attributed to the interventions themselves. The PA test employed during the intervention required participants to push their legs against the plinth to perform hip extension movements. This likely involved the gluteus maximus, a key hip extensor, alongside the hamstrings, potentially influencing the outcomes of neuromuscular electrical stimulation (NMES) and proprioceptive neuromuscular facilitation (PNF). Future research should aim to refine measurement tools and explore alternative positions that promote more isolated activation of the hamstrings while also examining the long-term effects of crPNF and NMES on performance parameters and injury prevention.

## 5. Conclusions

In conclusion, both crPNF and crPNF + NMES protocols effectively enhanced hamstring flexibility in youth basketball players in the short term, without producing detrimental effects on vertical jump performance. While NMES did not provide additional benefits in this study, its potential role in enhancing PNF effectiveness warrants further investigation.

## Figures and Tables

**Figure 1 jfmk-09-00280-f001:**
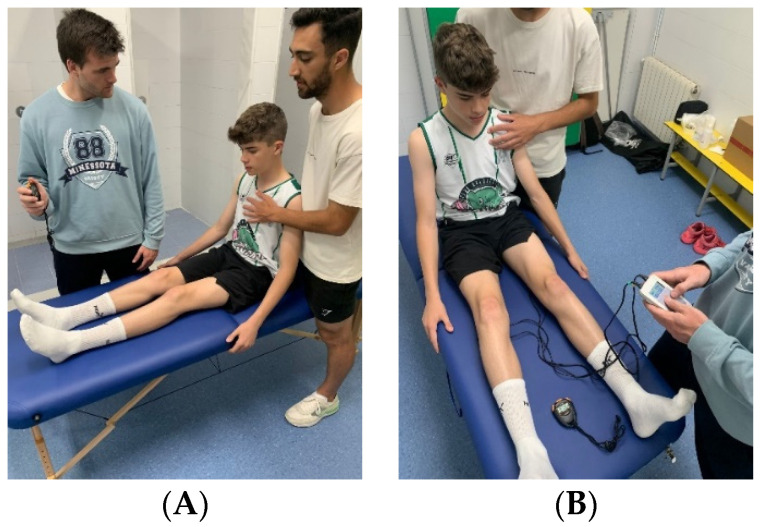
Intervention protocols. (**A**), intervention for proprioceptive neuromuscular facilitation (crPNF) group; (**B**), intervention for neuromuscular electrical stimulation (NMES) group.

**Figure 2 jfmk-09-00280-f002:**
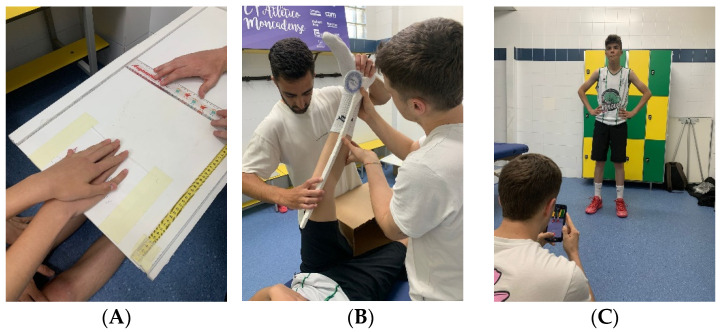
Variable Measurement: (**A**) Sit and Reach test; (**B**) popliteal angle test; (**C**) Counter Movement Jump test.

**Figure 3 jfmk-09-00280-f003:**
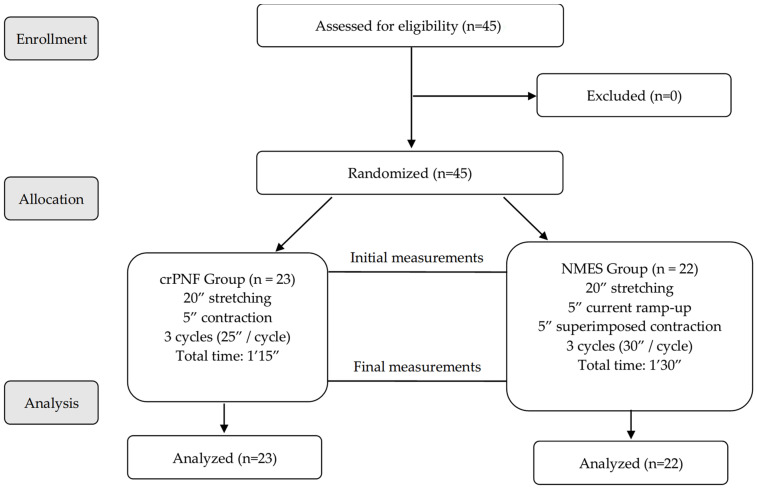
Flow diagram showing the phases of the study.; Abbreviations: PNF, proprioceptive neuromuscular facilitation; NMES, neuromuscular electrical stimulation.

**Table 1 jfmk-09-00280-t001:** Demographic characteristics of each intervention group, expressed as mean (Standard Deviation).

	PNF Group (*n* = 23)	NMES Group (*n* = 22)
Age	14.2 (1.4)	14.6 (1.6)
Height (cm)	170.6 (11.9)	172.8 (10.1)
Weight (kg)	58.2 (12.7)	61.5 (12.6)
Arm Span (cm)	172.6 (13.2)	174.0 (10.4)
Height–Arm Span (cm)	−2.0 (4.6)	−1.2 (3.8)

Abbreviations: PNF, proprioceptive neuromuscular facilitation; NMES, neuromuscular electrical stimulation.

**Table 2 jfmk-09-00280-t002:** Changes in Sit-and-Reach, Popliteal Angle, and Countermovement Jump Outcomes Pre- and Post-Intervention in crPNF and NMES Groups.

	Group									
	crPNF (*n* = 23)					NMES (*n* = 22)				
	Pre	Post	Difference (95% CI)	Cohen’s d	*p*-Value	Pre	Post	Difference (95% CI)	Cohen’s d	*p*-Value
SR (cm)	31.5 (6.5)	34.5 (6.6)	3 (2.2, 3.8)	2.307	<0.001 **	31.1 (10.2)	34.4 (9.4)	3.3 (2.5, 4.2)	2.496	<0.001 **
PA (°)	28.8 (11.8)	23.6 (11.8)	−5.2 (−7.1, −3.3)	1.733	<0.001 **	27.9 (12.1)	22.3 (12.3)	−5.7 (−7.5, −3.8)	1.831	<0.001 **
CMJ (cm)	32.8 (6.3)	33.0 (6.1)	0.2 (−0.4, 0.8)	0.222	0.471	29.7 (6.3)	29.5 (6.7)	−0.2 (−0.8, −0.4)	0.167	0.580

Abbreviations: PNF, proprioceptive neuromuscular facilitation; NMES, neuromuscular electrical stimulation; SR, Sit and Reach test; PA, popliteal angle test; CMJ, Counter Movement Jump. ** *p* < 0.001.

## Data Availability

The data presented in this study are available on request from the corresponding author. The data are not publicly available due to privacy concerns.

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
