# Peer review of "Short Term Effects of Proprioceptive Neuromuscular Facilitation Combined with Neuromuscular Electrical Stimulation in Youth Basketball Players: A Randomized Controlled Trial"

_jfmk, 2024, doi:10.3390/jfmk9040280_

Round 1
Reviewer 1 Report
Comments and Suggestions for Authors
The manuscript is well written, and the conclusions are based on the findings of your study.
The purpose of the study was to examine hamstring flexibility and jumping ability with crPNF and crPNF + NMES interventions in young basketball players using a single-blind randomized clinical trial. The pre and post effects of interventions on hamstring flexibility was evaluated with Sit & Reach Test (SR) and the Popliteal Angle Test (PA), and jumping ability was assessed with the Counter Movement Jump test (CMJ).
The topic that the authors addressed is relevant to the field and provides some additional information regarding the effects of crPNF and crPNF + NMES techniques on enhancing hamstring flexibility in young male basketball players, without affecting vertical jump ability. The methodology, the statistical analyses, tables and figures, as well as references, are appropriate for this research design.
This research study, while straightforward in its scope, is presented in a well-organized manner. Upon thorough review, I found no issues or areas requiring further revision.
Based on my expertise, my professional knowledge and my research interests, these are the points I would like to make, and I am confident that the manuscript meets the requirements for publication in the JFMK.
Author Response
Comment:
The manuscript is well written, and the conclusions are based on the findings of your study. The purpose of the study was to examine hamstring flexibility and jumping ability with crPNF and crPNF + NMES interventions in young basketball players using a single-blind randomized clinical trial. The pre and post effects of interventions on hamstring flexibility was evaluated with Sit & Reach Test (SR) and the Popliteal Angle Test (PA), and jumping ability was assessed with the Counter Movement Jump test (CMJ). The topic that the authors addressed is relevant to the field and provides some additional information regarding the effects of crPNF and crPNF + NMES techniques on enhancing hamstring flexibility in young male basketball players, without affecting vertical jump ability. The methodology, the statistical analyses, tables and figures, as well as references, are appropriate for this research design. This research study, while straightforward in its scope, is presented in a well-organized manner. Upon thorough review, I found no issues or areas requiring further revision. Based on my expertise, my professional knowledge and my research interests, these are the points I would like to make, and I am confident that the manuscript meets the requirements for publication in the JFMK.
Response:
We sincerely appreciate the time and effort you dedicated to reviewing our manuscript. Your kind and thorough evaluation of our work is greatly valued. Thank you very much.
I have attached a Word document with the same information, should it be necessary. Thank you very much.
Reviewer 2 Report
Comments and Suggestions for Authors
Dear authors,
I enjoyed reading your study, and this type of study will certainly find its audience. However, obtaining more relevant research results still requires a larger number of more homogeneous subjects. Also, the manuscript should be better structured, i.e., differently structured for more effortless reading so that the research can be applied to other research.
Title
First, the title states that it is a randomized clinical trial, which, in my opinion, is incorrect because randomized clinical trials assess medical interventions' effectiveness and safety, which is not the case in this research. Perhaps it would be more appropriate to call it a randomized controlled trial.
Introduction
The introduction is written quite briefly, is challenging to read, and does not introduce us in detail regarding the research issues. For example, the introduction does not describe the game of basketball or methods for developing flexibility, which is crucial for a better understanding of the research subject. Be sure to explain why you chose the mentioned stretching techniques compared to other techniques, i.e., what happens to the muscles to improve flexibility. Also, describe the NMES method in much more detail because if we use it, we should present it to the readers in an understandable way, as well as explain what happens to the muscles and what the difference is between it and other types of stretching.
The sentence in lines 48-50 should be clarified and better connected to the previous and following text.
Materials and methods
In the Study design, you mention that this study is a single-blind randomized clinical study, which it is not. Single-blind would mean that one group does not know what they are doing, i.e., which drugs they are receiving, which is not the case in this study. According to the Chapter Randomizing and Blinding, only the researcher is blinded (which is good because it cannot influence the results of the study).
In the chapter Subjects, you give us a modest description of the subjects. You do not even tell us how large the sample is or what its structure is (age, training age, gender, basic anthropological characteristics, etc.).
The chapters Participant and Interventions should also be merged for better text readability.
You determined the groups according to the PA test results, and you did not explain anywhere why you used the PA results instead of the SR test, which is also intended to measure the flexibility of the posterior compartment.
You also used the average of the PA test from three attempts rather than the maximum value. Explain this in the text.
In CMJ, you did not write what device you used to measure the jump height and where their hands were during the test, which is very important for repeated research.
Results
In the results section, you describe the sample of respondents and the time of conducting the tests, which should be in the materials and methods. I would like you to move this to the mentioned section.
Also, the results of the Shapiro-Wilk test will be moved to the statistical analysis section.
Discussion
Considering the obtained research results, the discussion is well-written
Author Response
Comment 1: Dear authors,
-I enjoyed reading your study, and this type of study will certainly find its audience. However, obtaining more relevant research results still requires a larger number of more homogeneous subjects. Also, the manuscript should be better structured, i.e., differently structured for more effortless reading so that the research can be applied to other research.
Response 1: We sincerely appreciate the time and effort you dedicated to reviewing our manuscript. Your thoughtful feedback is invaluable, and we firmly believe that it, combined with input from the other reviewers, has significantly improved the quality and clarity of the paper. In response to your suggestions, we have restructured the manuscript to enhance its readability and facilitate its application to future research. Below, we provide a detailed, point-by-point response to each of your comments. Thank you once again for your important contributions.
Title
Comment 2: -First, the title states that it is a randomized clinical trial, which, in my opinion, is incorrect because randomized clinical trials assess medical interventions' effectiveness and safety, which is not the case in this research. Perhaps it would be more appropriate to call it a randomized controlled trial.
Response 2: We fully agree with your observation regarding the title. Following your suggestion, we have revised it to 'randomized controlled trial' to better reflect the nature of the study. Thank you for pointing this out.
Introduction
Comment 3: -The introduction is written quite briefly, is challenging to read, and does not introduce us in detail regarding the research issues. For example, the introduction does not describe the game of basketball or methods for developing flexibility, which is crucial for a better understanding of the research subject. Be sure to explain why you chose the mentioned stretching techniques compared to other techniques, i.e., what happens to the muscles to improve flexibility. Also, describe the NMES method in much more detail because if we use it, we should present it to the readers in an understandable way, as well as explain what happens to the muscles and what the difference is between it and other types of stretching.
The sentence in lines 48-50 should be clarified and better connected to the previous and following text.
Response 3: Thank you for your detailed feedback on the introduction. Incorporating your valuable suggestions, as well as those of another reviewer, we have substantially revised this section to provide a comprehensive and accessible overview of the research topic. The introduction now includes a detailed description of basketball, focusing on its physical demands and the critical role of flexibility and vertical jump performance in achieving success. We have also addressed the prevalence and implications of hamstring tightness, particularly in male athletes, and its relevance to basketball players, including its association with injury risks.
In response to your comments, we have expanded the explanation of the selected stretching techniques, providing a detailed rationale for why PNF was chosen over other methods. The description of PNF, including its contract-relax variant (crPNF), has been enriched to highlight its mechanisms of action and effectiveness in improving flexibility. Similarly, we have elaborated on the NMES method, detailing its physiological effects on muscles, such as its influence on muscle contractions, the H-reflex, and pain modulation, and explaining how it differs from other stretching techniques. These changes have significantly improved the introduction, making it more informative, logically structured, and easier to read, while providing a stronger context for the study. Thank you again for your insightful suggestions, which have greatly contributed to enhancing the manuscript.
Materials and methods
Comment 4: -In the Study design, you mention that this study is a single-blind randomized clinical study, which it is not. Single-blind would mean that one group does not know what they are doing, i.e., which drugs they are receiving, which is not the case in this study. According to the Chapter Randomizing and Blinding, only the researcher is blinded (which is good because it cannot influence the results of the study).
Response 4: Thank you for your observation regarding the study design. You are absolutely right that the term 'single-blind randomized clinical study' is not the most appropriate, as it is generally associated with studies involving medical interventions such as drug trials. In this case, only the researcher conducting the measurements was blinded, which is consistent with the methodology outlined in the Chapter 'Randomizing and Blinding.' We have revised the manuscript to accurately reflect this and appreciate you bringing it to our attention.
Comment 5: -In the chapter Subjects, you give us a modest description of the subjects. You do not even tell us how large the sample is or what its structure is (age, training age, gender, basic anthropological characteristics, etc.).
Response 5: Thank you for your observation regarding the description of the subjects in the 'Participants' section. In line with the CONSORT statement for RCTs, detailed information about the sample size, age, gender, and basic anthropometric characteristics is already included in the Results section (Table 1). However, to enhance clarity, we have also added the following sentence to the 'Participants' section: “A total of 45 participants were recruited between April 15 and April 28, 2024, and the measurements of the variables were conducted between April 30 and May 8, 2024.”
We understand the importance of presenting a comprehensive description earlier in the manuscript. If you believe further adjustments are necessary, we would be happy to revise the text in the 'Participants' section for greater clarity.
Comment 6: -The chapters Participant and Interventions should also be merged for better text readability.
Response 6: Thank you for your suggestion regarding merging the 'Participants' and 'Interventions' sections. However, we have kept these sections separate to align with the CONSORT guidelines, which recommend providing detailed information about the eligibility criteria and recruitment process in the 'Participants' section (item 4) and describing the interventions for each group in a distinct section (item 5). This structure ensures clarity and allows readers to easily locate specific details about the study's design and methodology. We believe maintaining this separation enhances the manuscript's readability and adheres to established reporting standards.
Comment 7: -You determined the groups according to the PA test results, and you did not explain anywhere why you used the PA results instead of the SR test, which is also intended to measure the flexibility of the posterior compartment.
Response 7: Thank you for bringing up this point. We selected the PA test results for group stratification because this test is a validated and widely recognized tool for assessing hamstring flexibility. It provides precise angular measurements that are particularly well-suited for identifying degrees of muscle shortening. In contrast, while the SR test is a valuable measure of posterior chain flexibility, it incorporates additional factors, such as spinal and hip mobility, which could introduce variability unrelated to hamstring extensibility. To address your observation, we have clarified this rationale in the manuscript, as reflected in the revised paragraph:
“Participants were randomized into the two intervention groups using the stratification method. Based on the PA variable results, participants were classified according to their degree of hamstring shortening: normal (≤ 15°), grade I shortening (16°-34°), and grade II shortening (≥ 35°); following the reference values established by the literature [32]. The PA test was chosen for stratification because it provides precise angular measurements specific to hamstring extensibility, minimizing the influence of variables unrelated to muscle flexibility, such as spinal or hip mobility, which could affect the results of the SR test.”
Comment 8: -You also used the average of the PA test from three attempts rather than the maximum value. Explain this in the text.
Response 8: Thank you for your valuable comment, which we greatly appreciate. At the request of another reviewer, we have performed and included in the manuscript a reliability analysis focusing on relative reliability (ICC), yielding excellent results for all three variables (ICC > 0.96, p<0.001). We have also calculated the relative reliability (ICC) and MDC95 values and included all of them in the Materials and Methods and Results sections of the article. Additionally, we repeated the statistical analyses using the maximum values from the three attempts for all variables (PA, SR, and CMJ), confirming that the results remain consistent with those obtained using the average values.
Comment 9: -In CMJ, you did not write what device you used to measure the jump height and where their hands were during the test, which is very important for repeated research.
Response 9: Thank you again for your suggestion; it is indeed relevant. We have clarified all this information in the text to ensure greater transparency and reproducibility.
Results
Comment 10: -In the results section, you describe the sample of respondents and the time of conducting the tests, which should be in the materials and methods. I would like you to move this to the mentioned section.
Response 10: Thank you for your suggestion. We agree that the time of conducting the tests is more appropriately placed in the Materials and Methods section. We have moved this information accordingly to ensure consistency and improve the structure of the manuscript.
Comment 11: -Also, the results of the Shapiro-Wilk test will be moved to the statistical analysis section.
Response 11: Thank you for your suggestion. The results of the Shapiro-Wilk test have been moved to the Statistical Analysis section as recommended.
Discussion
Comment 12: -Considering the obtained research results, the discussion is well-written.
Response 12: Thank you for your positive feedback regarding the discussion section. We are glad that you found it well-written and aligned with the research results. Your comments are greatly appreciated.
I have attached a Word document with the same information, should it be necessary. Thank you very much.
Reviewer 3 Report
Comments and Suggestions for Authors
First of all, I would like to congratulate the authors for such an interesting study in the field of health.
It is clear that the Consort Guide is followed, including the design in the title. In addition, the title is very clear about the objective of the article.
The article is well structured. The introduction and objectives are clear. It is recommended to indicate the sample in the method section instead of the results.
The introduction is well laid out and presents a good guiding thread for the reader.
The objective is not very clear as it is written. It is recommended to improve it.
It is recommended to include one or more hypotheses.
The authors are congratulated for having registered with Clinical Trial, presenting the ethics committee as well as the compliance with the Declaration of Helsinki.
The inclusion and exclusion criteria, as well as the description of the recruitment of volunteers, are well detailed and explained. Even so, it would be appreciated if it were indicated at what level of professionalism the research was carried out.
The authors indicate that informed consent was taken from the parents or legal guardians of the participants. Even so, the participant's consent should be recorded. Was it not collected?
The applied intervention protocols are described. However, they should be detailed in greater depth, as well as including citations from research that has applied these protocols or that supports their implementation.
It is necessary to describe the conditions under which the volunteer came to the measurements. If a warm-up had been performed beforehand or not.
It is recommended to always use the same order in the variables (test), sit and reach, popliteal angle and vertical jump.
The groups have been randomized taking into account the degree of shortness of the subjects. The authors are congratulated. It is suggested that it could be interesting to analyze the results taking this variable into account.
Author Response
Comment 1: -First of all, I would like to congratulate the authors for such an interesting study in the field of health.
It is clear that the Consort Guide is followed, including the design in the title. In addition, the title is very clear about the objective of the article.
The article is well structured. The introduction and objectives are clear.
Response 1: Thank you very much for your kind and encouraging comments. We deeply appreciate your recognition of the study's relevance and your positive feedback regarding the clarity of the title, objectives, and adherence to the CONSORT guidelines. Your remarks are truly motivating and reinforce our efforts to ensure a well-structured and impactful manuscript.
Comment 2: -It is recommended to indicate the sample in the method section instead of the results.
Response 2: Thank you for your observation regarding the sample description in the 'Participants' (Methods) section. To enhance clarity, we have added the following sentence to the 'Participants' section: “A total of 45 participants were recruited between April 15 and April 28, 2024, and the measurements of the variables were conducted between April 30 and May 8, 2024.” In line with the CONSORT statement for RCTs, detailed information about the sample size, age, gender, and basic anthropometric characteristics is already included in the Results section (Table 1).
Comment 3: -The introduction is well laid out and presents a good guiding thread for the reader. The objective is not very clear as it is written. It is recommended to improve it. It is recommended to include one or more hypotheses.
Response 3: Thank you for your recommendation. We have clarified the objective to improve the focus and coherence of the manuscript as per your recommendation. We appreciate your suggestion. Additionally, based on suggestions from other reviewers, we have further elaborated on certain concepts, such as stretching techniques and the NMES method.
Comment 4: -The authors are congratulated for having registered with Clinical Trial, presenting the ethics committee as well as the compliance with the Declaration of Helsinki.
Response 4: Thank you for your comment. We ensured compliance with formal requirements such as trial registration, ethical approval, and adherence to the Declaration of Helsinki, and we appreciate your acknowledgment of this.
Comment 5: -The inclusion and exclusion criteria, as well as the description of the recruitment of volunteers, are well detailed and explained. Even so, it would be appreciated if it were indicated at what level of professionalism the research was carried out.
Response 5: Thanks for the suggestion; it is a good point. The following sentence has been added: “The participants competed in regional leagues organized by the Federación de Baloncesto de la Comunidad Valenciana (FBCV).
Comment 6: -The authors indicate that informed consent was taken from the parents or legal guardians of the participants. Even so, the participant's consent should be recorded. Was it not collected?
Response 6: You are absolutely correct, and we appreciate your observation. Participants' consent was collected along with parental or guardian consent. We have now included this information for clarity. Thank you for pointing it out.
Comment 7: -The applied intervention protocols are described. However, they should be detailed in greater depth, as well as including citations from research that has applied these protocols or that supports their implementation.
Response 7: Thank you once again for your suggestion. We have provided a more detailed description of the intervention protocols and added the relevant new citations to support their implementation.
Comment 8: -It is necessary to describe the conditions under which the volunteer came to the measurements. If a warm-up had been performed beforehand or not.
Response 8: Again, thank you for pointing this out, it is important. We realized this information was missing and have now included it in the manuscript. Specifically, we have added the following sentence:
“Prior to the measurements, all participants performed a standardized 10-minute warm-up to ensure consistency and reduce variability in the results. The warm-up included jogging, dynamic stretching, lower and upper limb strength exercises, submaximal plyometric exercises, and submaximal intermittent running with directional changes.”
Comment 9: -It is recommended to always use the same order in the variables (test), sit and reach, popliteal angle and vertical jump.
Response 9: Thank you for your suggestion. We have ensured that the same order— sit and reach (SR), popliteal angle (PA), and vertical jump (CMJ)—is consistently followed in the 'Variables and Measurements' section to maintain clarity and coherence. In addition, we have added the following sentence: “The tests were always conducted in the same order—SR, PA, and CMJ—to ensure consistency across participants, and enhance the reproducibility of the methodology.”
Comment 10: -The groups have been randomized taking into account the degree of shortness of the subjects. The authors are congratulated. It is suggested that it could be interesting to analyze the results taking this variable into account.
Response 10: Thank you again for your suggestion; it is certainly pertinent. As we also mentioned in response to another reviewer who proposed performing a one-way repeated measures ANCOVA with the pre-test as a covariate, this is a randomized controlled trial, and the randomization process was designed to balance baseline characteristics, including the degree of shortness, across groups. Our pre-test analysis confirmed that there were no statistically significant differences between groups at baseline, further supporting the appropriateness of the two-way repeated measures ANOVA for analyzing the data.
I have attached a Word document with the same information, should it be necessary. Thank you very much.
Reviewer 4 Report
Comments and Suggestions for Authors
General comments:
This study replicated previous similar studies using the young/adolescent male basketball players as the subjects. Without identifying some important research gaps or problems, the significance and novelty of this study are somewhat limited
The introduction was not able to set up good foundation science and theory behind to justify the need and importance of PNF or PNF+NMES. There was also confusion regarding the hamstring shortening of athletic population in leading to the need of this study to compare the effects to previous published studies (general population) on hamstring flexibility enhancement. However, the study design did not investigate the effect of hamstring shortening (baseline shortening/tightness) on the improvement difference between two protocols.
In the experimental design and data collection, many procedures were vaguely reported or not strictly controlled leading to many confounding variables or potential error (e.g. stretching position and testing)
Moreover, the study design itself did not include a true control group (no any intervention) for comparison and this is a very big limitation. Since repeated/multiple trials of 90/90 knee extension hamstring length test can already induce some warm up effects and improvement between pre-post comparison, without the implementation of the control group, the source of observable pre-post difference is questionable. Furthermore, the unequal treatment time between two protocols further exacerbate or magnify the potential errors/biases while these were not well mentioned.
The manuscript writing is kind of readable but not very well written. Some colloquial style and clumsy presentation decrease the readability somewhat.
Specific comments:
Topic: Young players? Please state clearly what young players
Introduction
Line 37 – 41: To say one of the main techniques to reduce the risk of hamstring strains is stretching…..it is dogmatic since stretching for flexibility enhancement to prevent hamstring injury is inconclusive while only limited evidence can support this. Even stretching for flexibility enhancement “may” prevent hamstring strain, the underlying principles that potentially underpin the stretching practice is about the shifting of length-torque curve or reduce the shortened optimum muscle length. Please double check all these papers
- Liu et al., 2012 https://doi.org/10.1016/j.jshs.2012.07.003
- Raya-Gonzalez et al., 2021 - 10.23736/S0022-4707.21.11670-6
- Ruan et al., 2018 - 10.1519/JSC.0000000000002645
Please elaborate the principles behind to support why flexibility training such as PNF can be one of the potential techniques (prefer to say potential techniques to tune down the tone rather than making this modality as well supported). This is important as if stretching for flexibility is not justified here, this study will have no value and function
Line 45 – a bit confusing…..may consider changing to “alternating stretching and voluntary isometric contraction for several times….”
Line 48-52 – I think it should be another paragraph as it sounds very weird to suddenly change from the introduction of PNF technique to basketball jumping things. Please add necessary sentences to help transit and smooth it out.
Line 50-53 – regarding the controversy…this part is not well written or does not read well. Few points should be highlighted…..Although some studies propose the possibility of decrease the jumping or power performance after prolonged stretching, the performance change is trivial if stretching time per muscle group < 60s and in addition, dynamic warm-up after the stretching / flexibility modalities seems to be able to further dilute the adverse effect of stretching and therefore, stretching can still be considered as appropriate option in warm-up section if flexibility or mobility enhancement is at the top priority
Line 54 to 68 – it is the main introduction to justify why PNF + NMES, however some important information is missing meanwhile this part is a bit jumpy and not structured. The author has lightly mentioned the crPNF +NMES applied in few previous studies and the method superimposed NMES was introduced. Please elaborate the underlying mechanisms/principles of superimposed NMES on muscle contraction and HOW this may further boost/optimize the original PNF performance (is it stronger muscle contraction due to superimposed potentially leading to greater inhibitory effect on H-reflex and muscle contractibility of the agonist?) ……or after using the NMES with certain parameters, there may be potentially some pain modulation and hence enhancing the pain tolerance and PNF improvement potential?? Please go deep to elaborate the principles behind in your introduction. These are important physiological theory foundation and science to support such a novel use of crPNF + NMES
Line 60 to 66 – a bit confusing or contradicted. In line 66 the idea of male subjects with higher degrees of hamstring shortening was mentioned. Does it mean young basketball players MUST have higher degrees of hamstring shortening? Regarding shortening, are you talking about relatively poorer flexibility or more frequent muscle shortening due to practice/competition? In line 62, the author mentioned the varies hamstring flexibility in different populations. Please show evidence that basketball players do have poorer flexibility than non-athlete population. If there is no such thing, please rewrite this part
Line 67 or 69, it may be considered as another separate paragraph as the last part of the introduction. Line 69 to 72 a bit clumsy that the author can simply say to compare the effects of crPNF and the combined crPNF + NMES techniques on hamstring flexibility
Line 72 to 73 – I don’t prefer to say “if either of the stretches negatively affects it”. It sounds like the authors have already had underlying bias or belief that crPNF or combined protocol likely alter the jumping performance in a negative manner. Moreover, this writing style more colloquial style rather than academic style, please rewrite.
Methods:
- Line 81 to 87 – in your introduction, you proposed that those young basketball players have higher degree of shortening and this is also your part of the study purpose. However, I don’t see that the baseline hamstring flexibility 90/90 test results were used for effects comparison. Moreover, the use of subjects between age 11 to 18 may come across pre-PHV, circa-PHV and post-PHV subjects. Those subjects at post-PHV I doubt they had a substantially poorer hamstring flexibility due to sudden increase of body height. Moreover, how many training experience do you need? At least 1 year or what? How many subjects were recruited per each group? Although you have mentioned in the results, it is more preferrable to have subjects’ information in method: subject section
Line 92 to 94 – why using long sitting for stretching?? Was there any previous study of PNF stretch using this position? Long sitting position is similar to sit and reach but it is NOT isolated hamstring stretch. It can be back extensor with unstable pelvic tilting during the stretching implementation. Regarding the contraction, how the subject perform the isometric contraction? Was he resisting against the plinth/massage table or resist again the manual resistance by the researcher? How much force applied in the isometric contraction (20, 60 or 100% of MVIC?)
Line 101 to 108 – regarding the NMES protocol, any citation from previous studies to support? Or just a novel protocol based on intuitive feeling?
Line 110 to 115 – what is the flow of testing? Which test first and which later?
Line 116 to 122 – the hamstring 90/90 assessment procedure should be more consistent and standardized according to previous studies. I think some recent studies using this test, the hip flexion fixation was accomplished by certain device instead of manual handling by one of the researchers. Sine the repeated 3 trials of this test itself already induce stretching and warm up effects to the muscle, therefore it is better to state clearly how many seconds have been taken place (such as within 2 seconds per reading or per trial).
Line 126 to 132 – from the photo, the sit and reach test setup was not very nice as the use of measurement tape putting on a table rather than using a standard sit and reach box. I don’t know if there is any reaching height difference when compared to the standard sit-and-reach box, leading to different flexibility results
Line 133 to 137 – the device e.g. iphone and what IOS should be stated clearly as the different smart phone version have different sampling rate (30 vs. 120) affecting the frequency when reading the take-off and landing time.
Line 116 to 137 – when saying high reliability, please state the ICC value at least using results from previous studies
Any warm up or familiarization session before conducting 3 tests?
Results:
Since 3 trials were performed for each test, the relative reliability ICC and absolute reliability SEM and MDC95 values should be given to understand the true changes
Figure 3 should be at the method section I think
Line 167 – the use of two-way repeated measures ANOVA can be an option but use of one-way repeated measure ANCOVA to treat the pre-test as covariate may be an even more accurate method. Moreover, did you calculate the effect size for the post hoc pairwise comparison?
Table 2 does not show all the important / essential information now – you should include:
- The difference after post-pre
- The *symbol showing p<0.05
- The effect size (Cohen’s d values?)
- The 95% CI of the mean difference
- The F value may not be necessary
In table 2, the AP crPNF Post = 11,. ??
Discussion:
Line 216 to 217 – the main mechanism for crPNF was not well proved yet. It could be just the modulation of pain tolerance
Line 222 to 224 – too dogmatic to say PNF can be used to address the high incidence of injuries. This is very misleading. To address/prevent incidence of hamstring strain, since the onset of injuries can be caused by multiple factors integrated together (modifiable factors such as eccentric strength, flexibility, pelvic control…and non-modifiable factors such as age, gender…etc.), therefore we cannot say simply applying PNF for increasing hamstring flexibility to address or prevent high injury incidence. Therefore line 226 to 229 are also misleading and invalid as PNF stretching alone or changing the PNF protocol is not supposed to be the most effective/impactful way leading to observable difference of injury prevalence especially when flexibility and stretching are still inconclusive for hamstring injury prevention
Line 241 to 243 – any citation to support the difference between unaccustomed and accustomed stimulus leading to higher electrical intensity and hence PNF effect?
Overall the discussion could not provide any conclusive findings and proposed reasons for explaining such findings are relatively weak and marginal. Without strictly controlling the data collection quality, protocol implementation and procedures, there were lots of potential error in different part of the data collection components finally leading to some untrustworthy results and inconclusive findings.
Comments on the Quality of English LanguageGenerally readable but the presentation is not in a very smart and efficient way.
Author Response
*General comments:
Comment 1: This study replicated previous similar studies using the young/adolescent male basketball players as the subjects. Without identifying some important research gaps or problems, the significance and novelty of this study are somewhat limited
The introduction was not able to set up good foundation science and theory behind to justify the need and importance of PNF or PNF+NMES. There was also confusion regarding the hamstring shortening of athletic population in leading to the need of this study to compare the effects to previous published studies (general population) on hamstring flexibility enhancement. However, the study design did not investigate the effect of hamstring shortening (baseline shortening/tightness) on the improvement difference between two protocols.
In the experimental design and data collection, many procedures were vaguely reported or not strictly controlled leading to many confounding variables or potential error (e.g. stretching position and testing).
Moreover, the study design itself did not include a true control group (no any intervention) for comparison and this is a very big limitation. Since repeated/multiple trials of 90/90 knee extension hamstring length test can already induce some warm up effects and improvement between pre-post comparison, without the implementation of the control group, the source of observable pre-post difference is questionable. Furthermore, the unequal treatment time between two protocols further exacerbate or magnify the potential errors/biases while these were not well mentioned.
The manuscript writing is kind of readable but not very well written. Some colloquial style and clumsy presentation decrease the readability somewhat.
Response 1: We sincerely appreciate the time and effort you dedicated to reviewing our manuscript. Your insightful feedback, along with the input from the other reviewers, has been invaluable in significantly enhancing the quality of our work. Thank you for your thoughtful contributions. Below, we provide a detailed, point-by-point response to each of your comments. The Introduction and Discussion sections have been thoroughly revised to address your general comments. We believe these changes have significantly improved the manuscript. Thank you once for your valuable input!
*Specific comments:
Comment 2: -Topic: Young players? Please state clearly what young players
Response 2: Thank you for your observation. We have revised the title to specify the population studied. The updated title is: "Short Term Effects of Proprioceptive Neuromuscular Facilitation Combined with Neuromuscular Electrical Stimulation in Youth Basketball Players: A Randomized Controlled Clinical Trial."
Introduction
Comment 3: Line 37 – 41: To say one of the main techniques to reduce the risk of hamstring strains is stretching…..it is dogmatic since stretching for flexibility enhancement to prevent hamstring injury is inconclusive while only limited evidence can support this. Even stretching for flexibility enhancement “may” prevent hamstring strain, the underlying principles that potentially underpin the stretching practice is about the shifting of length-torque curve or reduce the shortened optimum muscle length. Please double check all these papers
- Liu et al., 2012 https://doi.org/10.1016/j.jshs.2012.07.003
- Raya-Gonzalez et al., 2021 - 10.23736/S0022-4707.21.11670-6
- Ruan et al., 2018 - 10.1519/JSC.0000000000002645
Please elaborate the principles behind to support why flexibility training such as PNF can be one of the potential techniques (prefer to say potential techniques to tune down the tone rather than making this modality as well supported). This is important as if stretching for flexibility is not justified here, this study will have no value and function
Line 45 – a bit confusing…..may consider changing to “alternating stretching and voluntary isometric contraction for several times….”
Line 48-52 – I think it should be another paragraph as it sounds very weird to suddenly change from the introduction of PNF technique to basketball jumping things. Please add necessary sentences to help transit and smooth it out.
Line 50-53 – regarding the controversy…this part is not well written or does not read well. Few points should be highlighted…..Although some studies propose the possibility of decrease the jumping or power performance after prolonged stretching, the performance change is trivial if stretching time per muscle group < 60s and in addition, dynamic warm-up after the stretching / flexibility modalities seems to be able to further dilute the adverse effect of stretching and therefore, stretching can still be considered as appropriate option in warm-up section if flexibility or mobility enhancement is at the top priority
Line 54 to 68 – it is the main introduction to justify why PNF + NMES, however some important information is missing meanwhile this part is a bit jumpy and not structured. The author has lightly mentioned the crPNF +NMES applied in few previous studies and the method superimposed NMES was introduced. Please elaborate the underlying mechanisms/principles of superimposed NMES on muscle contraction and HOW this may further boost/optimize the original PNF performance (is it stronger muscle contraction due to superimposed potentially leading to greater inhibitory effect on H-reflex and muscle contractibility of the agonist?) ……or after using the NMES with certain parameters, there may be potentially some pain modulation and hence enhancing the pain tolerance and PNF improvement potential?? Please go deep to elaborate the principles behind in your introduction. These are important physiological theory foundation and science to support such a novel use of crPNF + NMES
Line 60 to 66 – a bit confusing or contradicted. In line 66 the idea of male subjects with higher degrees of hamstring shortening was mentioned. Does it mean young basketball players MUST have higher degrees of hamstring shortening? Regarding shortening, are you talking about relatively poorer flexibility or more frequent muscle shortening due to practice/competition? In line 62, the author mentioned the varies hamstring flexibility in different populations. Please show evidence that basketball players do have poorer flexibility than non-athlete population. If there is no such thing, please rewrite this part
Line 67 or 69, it may be considered as another separate paragraph as the last part of the introduction. Line 69 to 72 a bit clumsy that the author can simply say to compare the effects of crPNF and the combined crPNF + NMES techniques on hamstring flexibility
Line 72 to 73 – I don’t prefer to say “if either of the stretches negatively affects it”. It sounds like the authors have already had underlying bias or belief that crPNF or combined protocol likely alter the jumping performance in a negative manner. Moreover, this writing style more colloquial style rather than academic style, please rewrite.
Response 3: Thank you for your detailed and thoughtful feedback. In response to your comments, as well as those of other reviewers, we have completely revised the introduction to improve its structure, clarity, and comprehensiveness.
The revised introduction now provides a more detailed discussion of the research topic, including a thorough description of basketball as a sport, its physical demands, and the critical role of flexibility and vertical jump performance in athletic success. We have addressed the prevalence and implications of hamstring tightness, particularly in male athletes, and its relevance to basketball players, including its potential association with injury risks.
In addition to highlighting the potential role of flexibility training in reducing injury risks, we have emphasized its importance in enhancing athletic performance. Specifically, improved hamstring flexibility contributes to more efficient movement patterns, smoother execution of explosive actions such as sprints and vertical jumps, and greater overall biomechanical efficiency. This connection between flexibility and performance further underscores the value of the study.
We have also expanded on the rationale for selecting the stretching techniques used in the study, detailing the principles behind PNF and explaining why it was chosen over other methods. We elaborated on the mechanisms and effects of NMES, including its physiological influence on muscle contractions, the H-reflex, and pain modulation, and how these may enhance the effectiveness of PNF stretching.
We believe these changes have significantly improved the manuscript by making the introduction more informative, logically structured, and easier to follow, while providing a stronger scientific foundation for the study. Thank you again for your invaluable suggestions, which have greatly contributed to enhancing the quality of the paper.
Methods:
Comment 4: -Line 81 to 87 – in your introduction, you proposed that those young basketball players have higher degree of shortening and this is also your part of the study purpose. However, I don’t see that the baseline hamstring flexibility 90/90 test results were used for effects comparison. Moreover, the use of subjects between age 11 to 18 may come across pre-PHV, circa-PHV and post-PHV subjects. Those subjects at post-PHV I doubt they had a substantially poorer hamstring flexibility due to sudden increase of body height. Moreover, how many training experience do you need? At least 1 year or what? How many subjects were recruited per each group? Although you have mentioned in the results, it is more preferrable to have subjects’ information in method: subject section.
Response 4: As previously mentioned, the introduction has been completely revised to address your suggestions as well as those provided by other reviewers. We have also restructured the information related to the subjects. We sincerely appreciate your valuable feedback and the opportunity to improve the manuscript. Thank you very much.
Comment 5: -Line 92 to 94 – why using long sitting for stretching?? Was there any previous study of PNF stretch using this position? Long sitting position is similar to sit and reach but it is NOT isolated hamstring stretch. It can be back extensor with unstable pelvic tilting during the stretching implementation. Regarding the contraction, how the subject perform the isometric contraction? Was he resisting against the plinth/massage table or resist again the manual resistance by the researcher? How much force applied in the isometric contraction (20, 60 or 100% of MVIC?)
Response 5: Thank you for raising these points. We selected the long sitting position for its practicality and ease of application in this study. While we acknowledge that this position does not traditionally isolate the hamstrings, previous studies have supported its use in similar protocols (e.g., Pérez-Bellmunt et al.). To minimize the involvement of back extensors and pelvic tilting, participants were instructed to maintain a neutral spine position, which was carefully monitored by the researchers throughout the stretching and contraction phases. For the isometric contraction, participants pressed against the plinth/massage table, which provided a stable and consistent point of resistance. The contraction intensity was maximal voluntary effort (100% MVIC), as instructed to the participants, although the exact force applied was not quantified. This has been addressed in the manuscript as follows:
"Participants in the first group (crPNF Group) performed an isolated crPNF stretching protocol. They were placed in a long sitting position with maximum knee extension possible until a moderate-strong stretch sensation was felt, without pain. The stretch duration was 20 seconds, followed by a maximal voluntary isometric contraction (MVIC) of the hamstrings for 5 seconds (Pérez-Bellmunt). Three stretch-contraction cycles were completed. To minimize pelvic tilting and back extensor involvement, participants were instructed to maintain a neutral spine position, which was monitored by the researchers. Isometric contraction was performed by resisting against the plinth, providing a stable point of resistance. Participants were instructed to contract at maximal voluntary effort (100% MVIC). A second researcher controlled the stretching and contraction times (Figure 1A)."
We value your feedback and recognize the importance of specifying and measuring force levels, which will be addressed in future studies.
Comment 6: -Line 101 to 108 – regarding the NMES protocol, any citation from previous studies to support? Or just a novel protocol based on intuitive feeling?
Response 6: Thank you for the comment; this is an important point. We have now included two citations to support the protocol used, as follows:
“NMES superimposed during voluntary contractions is a proven method to enhance muscle performance without external loads (Labanca et al., 2022). The selected NMES parameters have been demonstrated to be effective and appropriate for this purpose (Labanca et al., 2022; Fukuda et al., 2013).”
The references are as follows:
- Labanca, L., Rocchi, J. E., Carta, N., Giannini, S., & Macaluso, A. (2022). NMES superimposed on movement is equally effective as heavy slow resistance training in patellar tendinopathy. Journal of Musculoskeletal & Neuronal Interactions, 22(4), 474.
- Fukuda, T. Y., Marcondes, F. B., dos Anjos Rabelo, N., de Vasconcelos, R. A., & Junior, C. C. (2013). Comparison of peak torque, intensity and discomfort generated by neuromuscular electrical stimulation of low and medium frequency. Isokinetics and Exercise Science, 21(2), 167-173.
Comment 7: -Line 110 to 115 – what is the flow of testing? Which test first and which later?
Response 7: Thank you for your observation; it highlights an important aspect of our methodology. We have clarified the testing flow in the manuscript. The tests were always conducted in the following order: sit and reach (SR), popliteal angle (PA), and vertical jump (CMJ). This sequence was maintained for all participants to ensure consistency and enhance the reproducibility of the methodology. We have included the following sentence in the 'Variables and Measurements' section to explicitly describe this:
"The tests were always conducted in the same order—SR, PA, and CMJ—to ensure consistency across participants and enhance reproducibility."
Comment 8: -Line 116 to 122 – the hamstring 90/90 assessment procedure should be more consistent and standardized according to previous studies. I think some recent studies using this test, the hip flexion fixation was accomplished by certain device instead of manual handling by one of the researchers. Since the repeated 3 trials of this test itself already induce stretching and warm up effects to the muscle, therefore it is better to state clearly how many seconds have been taken place (such as within 2 seconds per reading or per trial).
Response 8: Thank you for your insightful comment. We fully recognize the importance of ensuring consistency and standardization in the PA test. In our study, hip flexion was stabilized manually by one of the researchers to maintain uniformity across all participants. While we acknowledge that the use of a fixation device could enhance standardization, our approach was carefully controlled and consistent throughout the study. This is supported by the results of the reliability analysis, which we have now included in the manuscript based on your suggestion.
Regarding the potential stretching or warm-up effects of repeated trials, we agree that this is an important consideration. To address this, we ensured that each reading was taken within 2 seconds per trial, minimizing the duration of muscle engagement and reducing potential cumulative effects. This detail has been added to the Methods section to enhance transparency and reproducibility.
We sincerely appreciate your valuable feedback, which has helped us refine and improve the description of our methodology.
Comment 9: -Line 126 to 132 – from the photo, the sit and reach test setup was not very nice as the use of measurement tape putting on a table rather than using a standard sit and reach box. I don’t know if there is any reaching height difference when compared to the standard sit-and-reach box, leading to different flexibility results.
Response 9: Thank you for your observation. You are correct that the sit-and-reach test setup, as shown in the photo, used a measurement tape placed on a table rather than a standard sit-and-reach box. While this setup differs from the conventional one, it was consistently applied to all participants throughout the study to eliminate potential biases and ensure uniformity in the measurements.
We acknowledge that the table-based setup may result in slight differences compared to using a standard sit-and-reach box, particularly in reaching height. However, since the same procedure was followed for all subjects, we are confident that this approach does not compromise the internal validity of our findings. To address this, we have clarified this detail in the manuscript and acknowledged it as a potential limitation when comparing our results to studies using the standard box, as follows:
“The sit-and-reach test was performed using a measurement tape on a table instead of a standard sit-and-reach box. Although this setup ensured consistency across all participants and minimized potential bias, it may lead to slight differences when comparing our results to studies using the standard box.”
Comment 10: -Line 133 to 137 – the device e.g. iphone and what IOS should be stated clearly as the different smart phone version have different sampling rate (30 vs. 120) affecting the frequency when reading the take-off and landing time.
Response 10: Thank you again for your valuable comment. We have now included the specific details regarding the device and operating system used, as follows:
“The My Jump App, which has high reliability and validity for measuring vertical jump height in centimeters [GençoÄŸlu et al 2023], was used on an iPhone 11 running iOS 17.4.1.”
Comment 11: -Line 116 to 137 – when saying high reliability, please state the ICC value at least using results from previous studies
Response 11: Thank you for your observation. We have expanded the "Variables and Measurements" section to include additional references that support the reliability of the measurements. Additionally, as explained later in the manuscript, we have calculated the ICC for the variables measured in our study.
We appreciate your suggestion, as it strengthens the methodological rigor and clarity of the manuscript.
Comment 12: Any warm up or familiarization session before conducting 3 tests?
Response 12: Thank you for pointing this out. You are absolutely correct; this information was missing from the manuscript. We have now included it as follows:
“Prior to the measurements, all participants performed a standardized 10-minute warm-up to ensure consistency and reduce variability in the results. The warm-up included jogging, dynamic stretching, lower and upper limb strength exercises, submaximal plyometric exercises, and submaximal intermittent running with directional changes.”
Results:
Comment 13: -Since 3 trials were performed for each test, the relative reliability ICC and absolute reliability SEM and MDC95 values should be given to understand the true changes.
Response 13: Thank you for your suggestion. We have calculated the relative reliability (ICC), absolute reliability (SEM), and MDC95 values and included them in the Materials and Methods and Results sections of the article. The analysis revealed ICC values exceeding 0.96 for all three variables, with the following results: 0.975 (95% CI: 0.959–0.985) for SR, 0.990 (95% CI: 0.984–0.994) for PA, and 0.962 (95% CI: 0.939–0.978) for CMJ.
Comment 14: -Figure 3 should be at the method section I think
Response 14: Thanks again for your suggestion. Figure 3 corresponds to the flowchart of the randomized controlled trial, which illustrates the participant flow according to CONSORT guidelines. As such, we have included it in the Results section to present the actual recruitment and allocation outcomes. This placement follows standard reporting practices for RCTs. However, we are open to further discussion if you believe an alternative placement would improve clarity.
Comment 15: -Line 167 – the use of two-way repeated measures ANOVA can be an option but use of one-way repeated measure ANCOVA to treat the pre-test as covariate may be an even more accurate method. Moreover, did you calculate the effect size for the post hoc pairwise comparison?
Response 15: The two-way repeated measures ANOVA was chosen because it aligns with the design of our randomized controlled trial, which involves two independent factors: time (repeated measures) and groups. This method allows us to evaluate the main effects of time and group, as well as their interaction (time x group), which is crucial for understanding the differential impact of the interventions.
As this is a randomized controlled trial, the randomization process aimed to balance baseline characteristics between groups, including the pre-test values. Our pre-test analysis confirmed that there were no statistically significant differences between groups at baseline, supporting the appropriateness of the two-way repeated measures ANOVA in this context.
Regarding the effect size for post hoc pairwise comparisons, we agree on its importance and have now included these values (Partial eta2) in the revised manuscript to enhance the interpretation of our findings. This information has been also added to the Methods section for clarity and completeness. Thank you very much for your comment.
Comment 16: -Table 2 does not show all the important / essential information now – you should include:
The difference after post-pre
The *symbol showing p<0.05
The effect size (Cohen’s d values?)
The 95% CI of the mean difference
The F value may not be necessary
In table 2, the AP crPNF Post = 11,. ??
Response 16: We appreciate your detailed suggestions for improving Table 2. We have updated the table to include all the requested elements. Thank you very much!
Discussion:
Comment 17: Line 216 to 217 – the main mechanism for crPNF was not well proved yet. It could be just the modulation of pain tolerance
Line 222 to 224 – too dogmatic to say PNF can be used to address the high incidence of injuries. This is very misleading. To address/prevent incidence of hamstring strain, since the onset of injuries can be caused by multiple factors integrated together (modifiable factors such as eccentric strength, flexibility, pelvic control…and non-modifiable factors such as age, gender…etc.), therefore we cannot say simply applying PNF for increasing hamstring flexibility to address or prevent high injury incidence. Therefore line 226 to 229 are also misleading and invalid as PNF stretching alone or changing the PNF protocol is not supposed to be the most effective/impactful way leading to observable difference of injury prevalence especially when flexibility and stretching are still inconclusive for hamstring injury prevention
Line 241 to 243 – any citation to support the difference between unaccustomed and accustomed stimulus leading to higher electrical intensity and hence PNF effect?
Overall the discussion could not provide any conclusive findings and proposed reasons for explaining such findings are relatively weak and marginal. Without strictly controlling the data collection quality, protocol implementation and procedures, there were lots of potential error in different part of the data collection components finally leading to some untrustworthy results and inconclusive findings.
Comments on the Quality of English Language: Generally readable but the presentation is not in a very smart and efficient way.
Response 17: Thank you again for your comments. As previously mentioned, the discussion has been completely rewritten to address all of your feedback.
I have attached a Word document with the same information, should it be necessary. Thank you very much.
Round 2
Reviewer 2 Report
Comments and Suggestions for Authors
Dear authors,
I initially requested that the study be rejected, but after your reaction to my comments, I must humbly say that I was wrong and that I propose the study for acceptance. I have no objections to the scientific paper written this way and believe its quality has improved significantly. I appreciate your cooperation.
Best regards
Author Response
Comment: Dear authors,
I initially requested that the study be rejected, but after your reaction to my comments, I must humbly say that I was wrong and that I propose the study for acceptance. I have no objections to the scientific paper written this way and believe its quality has improved significantly. I appreciate your cooperation.
Best regards
Response: Dear Reviewer#2,
Thank you for your kind words and for reconsidering your assessment of our manuscript. We deeply appreciate your constructive feedback, which has greatly contributed to improving the quality of our study.
We are grateful for your cooperation and support.
Best regards
Reviewer 4 Report
Comments and Suggestions for Authors
Thanks for the revision. To be honest, the quality of this study is somewhat marginal (just at the borderline between major revision and rejection) as many potential flaws, biases or confounding variables existed that somewhat decreased the value of this study. But undeniably, your revised version has improved a lot in all parts.
Line 53-54: consider adding the concepts of synergistic dominance using this PMID 31440415. This can well support what you mentioned.
Line 66-67: I understand you are using contract-relax this term. Clinically and also in many papers/reference books, this can also be called "hold-relax" for isometric contraction while the contract-relax method is referred to isotonic. You may consider adding a ( ) as a note --> (this may be called "hold-relax" in some literature due to the confusing terminology)
Line 110 - delete the full stop, and shift the next sentence back to line 110
Line 166 - please state clearly the video frame per second (fps) and resolution adopted....e.g. 1080 60p
Table 2 - for the post hoc pairwise comparison, should it be the Cohen's d value or standardized mean difference rather than the Partial eta2 ?
The paragraph of limitation - your study did not adopt the 90/90 position resembling the active knee extension test to perform knee flexion (nearly a pure hamstring contraction with a very minimal amount of gastrocnemius involvement). The current long sitting position requires the subject to push the legs against the plinth to perform a hip extension motion. However, part of the loading was shared by another key hip extensor, gluteus maximus. Therefore, it is speculated that PNF using joint movements without isolated muscle contraction may affect/limit the effect of NMES. Please highlight this if possible.
The last part of the last sentence "but highlight the importance of integrating......" I wonder if this was highlighted by your findings.....and if you should delete this part meanwhile adding "without producing detrimental effects on vertical jump performance" after "improving flexibility"
Author Response
Comment 1: Thanks for the revision. To be honest, the quality of this study is somewhat marginal (just at the borderline between major revision and rejection) as many potential flaws, biases or confounding variables existed that somewhat decreased the value of this study. But undeniably, your revised version has improved a lot in all parts.
Response 1: Thank you again for your thoughtful and honest feedback. We truly appreciate your recognition of the improvements made in the revised version of our manuscript, which have been greatly influenced by your valuable comments, as well as those of the other reviewers.
We will carefully address the additional points you raised to further enhance the quality and rigor of the study. Your insights have been invaluable, and we are sincerely grateful for your guidance throughout this process.
Comment 2: -Line 53-54: consider adding the concepts of synergistic dominance using this PMID 31440415. This can well support what you mentioned.
Response 2: Thank you for the suggestion. We have incorporated the concept of synergistic dominance into the text as follows:
“Flexibility is a cornerstone of physical conditioning programs, as it enables tissues to adapt to stress, enhances movement efficiency, and may reduce injury risk, particularly in high-demand sports like basketball [16]. Specifically, improving hamstring flexibility supports smoother, more efficient movements by optimizing joint mechanics and muscle extensibility, which are essential for explosive actions such as sprints and rapid directional changes. Mechanisms underlying these benefits include shifting the muscle length-torque curve, addressing shortened optimum muscle length [17,18], and mitigating the effects of synergistic dominance, as well as mitigating the effects of synergistic dominance. The latter, as described by Buckthorpe et al. [19], refers to the compensatory activation of dominant muscles over weaker ones, which can lead to imbalances and inefficiencies in joint mechanics. These principles highlight the potential importance of hamstring flexibility in supporting athletic performance in basketball”.
Comment 3: -Line 66-67: I understand you are using contract-relax this term. Clinically and also in many papers/reference books, this can also be called "hold-relax" for isometric contraction while the contract-relax method is referred to isotonic. You may consider adding a ( ) as a note --> (this may be called "hold-relax" in some literature due to the confusing terminology).
Response 3: Thank you for pointing this out. We have included the clarification in the text to prevent potential confusion, as follows:
“Stretching techniques, including static stretching, dynamic stretching, and proprioceptive neuromuscular facilitation (PNF), are widely used to enhance flexibility and range of motion (ROM) [7,16]. Among these, PNF techniques are often favored for their superior effectiveness in increasing ROM [20,21]. PNF involves alternating stretching with voluntary isometric contractions of either agonist or antagonist muscles. When the agonist muscle contraction is incorporated, the technique is referred to as contract-relax PNF (crPNF) [21–23], although some sources use the term "hold-relax" to emphasize its isometric nature.”
Comment 4: -Line 110 - delete the full stop, and shift the next sentence back to line 110
Response 4: Thank you for identifying this issue. We have corrected the error as requested.
Comment 5: -Line 166 - please state clearly the video frame per second (fps) and resolution adopted....e.g. 1080 60p.
Response 5: Thank you for your observation. We confirm that the video captures were recorded using precisely the parameters you provided as an example. To ensure clarity, we have added the following text to the manuscript:
“The captures were made using video recorded at 1080p resolution and 60 frames per second (fps).”
Comment 6: -Table 2 - for the post hoc pairwise comparison, should it be the Cohen's d value or standardized mean difference rather than the Partial eta2 ?
Response 6: Thank you for your suggestion. We recognize that Cohen's d is an intuitive and widely recognized measure of effect size for post hoc pairwise comparisons. At the same time, we would like to point out that partial eta squared is a standard effect size reported in ANOVA analyses, including post hoc comparisons, as it reflects the proportion of variance explained by the factor of interest within the model. That said, we fully agree that Cohen's d offers a more direct and accessible interpretation of the magnitude of differences between specific groups.
In response to your suggestion, we have recalculated and updated the effect sizes in Table 2 to reflect Cohen's d values instead of partial eta squared. Additionally, we have updated the Statistical Analysis section in the Methods to specify the use of Cohen's d for post hoc analyses, ensuring clarity and consistency throughout the manuscript.
Comment 7: -The paragraph of limitation - your study did not adopt the 90/90 position resembling the active knee extension test to perform knee flexion (nearly a pure hamstring contraction with a very minimal amount of gastrocnemius involvement). The current long sitting position requires the subject to push the legs against the plinth to perform a hip extension motion. However, part of the loading was shared by another key hip extensor, gluteus maximus. Therefore, it is speculated that PNF using joint movements without isolated muscle contraction may affect/limit the effect of NMES. Please highlight this if possible.
Response 7: We have carefully considered your observation regarding the PA test methodology and have revised the limitations section of the manuscript as follows:
“The PA test employed during the intervention required participants to push their legs against the plinth to perform hip extension movements. This likely involved the gluteus maximus, a key hip extensor, alongside the hamstrings, potentially influencing the outcomes of neuromuscular electrical stimulation (NMES) and proprioceptive neuromuscular facilitation (PNF). Future research should aim to refine measurement tools and explore alternative positions that promote more isolated activation of the hamstrings while also examining the long-term effects of crPNF and NMES on performance parameters and injury prevention.”
Comment 8: -The last part of the last sentence "but highlight the importance of integrating......" I wonder if this was highlighted by your findings.....and if you should delete this part meanwhile adding "without producing detrimental effects on vertical jump performance" after "improving flexibility"
Response 8: Thank you for this observation. You are correct that our findings did not specifically address the integration of crPNF into a multifactorial approach for injury prevention. To ensure that the conclusions are fully supported by our data, we have revised the sentence as follows:
“In conclusion, both crPNF and crPNF + NMES protocols effectively enhanced hamstring flexibility in youth basketball players in the short term, without producing detrimental effects on vertical jump performance. While NMES did not provide additional benefits in this study, its potential role in enhancing PNF effectiveness warrants further investigation”.
We believe this revision better reflects the outcomes of our study and appreciate your suggestion to improve clarity.